# Genome-Wide Identification, Classification, and Expression Analysis of the HD-Zip Transcription Factor Family in Apple (*Malus domestica* Borkh.)

**DOI:** 10.3390/ijms23052632

**Published:** 2022-02-27

**Authors:** Kai Liu, Xiaolei Han, Zhaolin Liang, Jiadi Yan, Peihua Cong, Caixia Zhang

**Affiliations:** 1Research Institute of Pomology, Chinese Academy of Agricultural Sciences, Xingcheng 125100, China; liukai2429@163.com (K.L.); hanxiaolei@caas.cn (X.H.); lliangzhaolin@163.com (Z.L.); jiadiyan@163.com (J.Y.); congph@163.com (P.C.); 2Key Laboratory of Biology and Genetic Improvement of Horticultural Crops, Xingcheng 125100, China

**Keywords:** HD-Zip family, apple, transcription factor, gene expression

## Abstract

Homeodomain-leucine zipper (*HD-Zip*) family genes are considered to play an important role in plant growth and stress tolerance. However, a genome-wide analysis of *HD-Zip* genes in apples (*Malus domestica* Borkh.) has not been performed. We detected 48 *MdHDZ* genes in the apple genome, and categorized them into three subfamilies on the basis of phylogenetic analysis. The chromosomal locations, gene/protein structures, and physiological and biochemical properties of these genes were analyzed. Synteny analysis revealed that segmental duplications were key in the expansion of the apple HD-Zip family. According to an analysis of cis-regulatory elements and tissue-specific expression patterns, *MdHDZ* genes may be widely involved in the regulation of apple growth and tolerance to environmental stresses. Furthermore, the transcript levels of apple *HD-Zip* I and II genes were up-regulated in response to fungal treatments. Expression of apple *HD-Zip* Ⅲ genes was enhanced during adventitious bud regeneration. This suggested possible roles of these genes in regulating the apple response to fungal infection, as well as adventitious bud regeneration. The current results may help us to better understand the evolution and function of apple *HD-ZIP* genes, and thus facilitate further research on plant resistance to fungal infection and in vitro regeneration.

## 1. Introduction

During environmental adaptation, plants produce a series of responses at the cell to physiological level, and these responses are usually regulated by transcription factors (TFs) [1]. Homeobox (HB) genes, related to various growth and development processes, as well as stress responses, are considered key for idioplasm melioration in crops [2]. Each HB gene encodes a conserved 61 amino acid sequence known as the homeodomain (HD), which is responsible for sequence-specific DNA binding. In plants, KNOTTED1, which was isolated from maize (*Zea mays L.*), was the first HD-containing protein. According to the sequence differences and location of their HD domains, homology of the flanking sequences, and other correlative domains, HD-containing proteins were classified into six families, including HD-Zip (homeodomain-leucine-zipper), KNOX (KNOTTED1-like homeobox), PHD-Finger (homeodomain-finger), Bell (bell domain), WOX (Wuschel-related homeobox), and ZF-HD (zinc finger-homeodomain) [3]. HD-Zip, containing the HD and additional leucine zipper (LZ) elements, is ubiquitous in plants [1]. In many species, *HD-Zip* genes are clustered into four subfamilies—*HD-Zip* I, *HD-Zip* II, *HD-Zip* III, and *HD-Zip* IV—according to gene structure and function [4]. HD-Zip I proteins have multiple functions, including the regulation of stress tolerance [5,6] and organ development [1]. HD-Zip II proteins are master regulators of hormone signaling [7] and light response [8]. HD-Zip III proteins are involved in embryogenesis [9], apical meristem development [10], and auxin polarity transport during plant development [11]. The HD-Zip IV group is mainly involved in trichome formation [12], anthocyanin accumulation [13], and epidermal cell differentiation [14].

The apple (*Malus domestica* Borkh.), including over 30 primary species, is a major fruit crop in temperate regions [15]. As a result of its popularity, global apple production is increasing steadily, now second only to the banana [16]. Alternaria blotch, a fungal plant disease caused by the *Alternaria alternata* apple pathotype (AAAP), has greatly damaged apple production [17,18]. Breeding pathogen-resistant apple varieties could be the safest and most effective method to control this disease. However, the ambiguous molecular mechanisms of the apple immune response to AAAP, along with a relatively long breeding cycle, have resulted in a lack of an immune cultivar. Although numerous studies have revealed the role of *HD-Zip* genes in mitigating stresses, most have focused on their responses to abiotic stress, whereas responses to biotic stress and resistance mechanisms were seldom investigated. For instance, *HD-Zip* I genes *AtHB7* and *AtHB12*, involved in the regulation stomata closure, were strongly expressed after ABA treatment and drought [19]. Overexpression of the *HD-Zip* I gene *Oshox22* significantly improved tolerance to long-term NaCl stress in rice [20]. With an improvement in photosynthetic performance under salt stress, the drought tolerance of *MdHB-7* transgenic apple trees became significantly stronger than that of control plants [21]. Data available on the roles of HD-Zip proteins in pathogen defense are few; however, a functional study of several HD-Zip I and II subfamily members in dicotyledons showed that HD-Zip may play a role in the hormone-mediated regulation of biotic stress. Virus-induced *CaHDZ27* silencing decreased the expression of pepper resistance genes, thus increasing the susceptibility of pepper to *R. solanacearum* infection [22]. GhHB12 inhibits the expression of resistance genes *GhJAZ2* and *GhPR3* by directly binding to their promoter regions, thereby making *GhHB12* transgenic cotton more susceptible to *Verticillium dahlia* [23]. Moreover, many differentially expressed *HD-Zip* genes were detected in pear trees during an infection of *Alternaria alternata* [24].

Gene function analysis is a central pursuit in molecular biology [25]. Gene silencing and overexpression—the basic approaches used in this analysis—depend on genetic transformation. In vitro plant regeneration has remarkable potential for constructing a genetic transformation system, and is the basis of gene function analysis [26]. *Agrobacterium*-mediated transformation is commonly used to introduce novel genes into plants, and its genetic transformation efficiency is strongly related to plant regeneration abilities [27]. However, the very low genetic efficiency of apples severely hinders gene function analysis [28,29,30]. In apples, the study of the molecular mechanisms underlying regeneration is conducive to the improvement of adventitious bud regeneration from leaves in vitro, as well as the enhancement of genetic transformation efficiency. Many *HD-Zip* genes involved in plant regeneration have been identified. For instance, *HD-Zip* III can specifically bind to B-type ARRs to form a transcription complex to activate *WUS* and regulate adventitious shoot regeneration [31,32]. *ATHB8* can respond to auxin signaling and regulate the differentiation of procambial and cambial cells [10]. *AGO10* inhibits in vitro shoot regeneration via *miR165/166* repression [33].

Based on previous studies, we predicted that apple *HD-Zip* genes may be involved in the response to AAAP infection and adventitious bud regeneration. *HD-Zip* family genes have been comprehensively analyzed in *Arabidopsis* [1,9,12,34], soybeans [35], citrus [36], cucumbers [37], potatoes [38], wheat [39], and other species [40,41]. However, little is known about this family of genes in the apple genome. To develop a better understanding of the roles of *HD-Zip* genes in apples, we conducted a genome-wide search for apple *HD-Zip* genes. A total of 48 full-length *HD-Zip* genes were identified in the apple genome. The physicochemical properties, gene structure, chromosome distribution, evolutionary relationship, and tissue-specific expression profiles of these genes were analyzed through the comprehensive bioinformatics method. Furthermore, the expression profiles of the apple *HD-Zip* genes in response to AAAP infection and adventitious bud regeneration were examined through RNA-Seq. The results showed that the expression of apple *HD-Zip* I and II genes were up-regulated under AAAP infection, while the expression of apple *HD-Zip* Ⅲ genes was enhanced during adventitious bud regeneration. We further confirmed these expression changes via a qRT-PCR experiment, and the results were consistent with the RNA-Seq data which indicated that apple *HD-Zip* genes play a key regulatory role in apples’ response to fungal infection and adventitious bud regeneration. These results provide insights for a more comprehensive understanding of the function of the *HD-Zip* gene family in apples.

## 2. Results

### 2.1. Identification of MdHDZ Genes

Using the *HD-Zip* gene sequence of *Arabidopsis* as a reference, BLASTP was applied to search for possible *HD-Zip* genes in the apple genome (GDDH13v1.1) [42]. Approximately 85 candidate genes were isolated. The presence of conserved HD and LZ domains were further verified by SMART and NCBI–CDD (Appendix A). In total, 48 *HD-Zip* genes were detected; this number is greater than that identified in *Arabidopsis*, rice, wheat, millet, and pepper, but lower than that identified in maize and soybean (Appendix A). The 48 identified apple *HD-Zip* genes were named *MdHDZ1* to *MdHDZ48**,* on the basis of their positions on the chromosomes.

As illustrated in Table 1, the molecular weight of these 48 MdHDZs ranged from 18.34 kDa (MdHDZ32) to 93.08 kDa (MdHDZ24), and their lengths ranged from 154 (MdHDZ32) AAs to 852 (MdHDZ10) AAs. Moreover, the pI of these proteins ranged from 4.75 (MdHDZ15) to 9.52 (MdHDZ11).

### 2.2. Phylogenetic and Synteny Analyses of MdHDZ Genes

By constructing an unrooted phylogenetic tree, the evolutionary and phylogenetic relationships between 47 *Arabidopsis* HD-Zip proteins and 48 MdHDZs were analyzed. The apple HD-Zip proteins were further divided into three main subfamilies (I–III), with subfamily I having a maximum of 22 genes, and subfamily II having a minimum of eight genes (Figure 1a). This distribution is different from that of most species, in which HD-Zip proteins are divided into four groups. According to studies on the conserved domains of HD-Zip subfamily II proteins, these proteins generally have an N-terminal consensus sequence, along with the HD and LZ domains, in other species [43]. Here, only 7 of the 18 subfamily II members contained the N-terminal consensus sequence (Appendix A).

*MdHDZ* genes are distributed across all apple chromosomes; however, the distribution is not uniform (Figure 1). One gene is located on chromosomes 0, 3, 4, 5, 10, and 11; two on chromosome 12; three on chromosomes 1, 2, 6, 7, 9, 14, and 17; four on chromosomes 8 and 16; five on chromosome 15; and six on chromosome 13. A significantly higher number of genes is distributed at the proximal or distal ends of chromosomes than in the middle. Tandem and segmental duplication events were analyzed to investigate the evolutionary mechanism of MdHDZ family genes. Based on gene sequence homology, 10 *MdHDZ* genes (20.83%) formed five tandem duplication pairs, while 41 *MdHDZ* genes (85.42%) formed 44 segmental duplication pairs (Figure 1b and Appendix A). Interestingly, most gene pairs with duplication events belonged to the same subfamily in the phylogenetic tree.

### 2.3. Gene Structure and Conserved Motif Analyses of MdHDZ Genes

The gene structure and conserved motif compositions of apple *HD-Zip* genes have been analyzed to determine their structural diversity. In addition, to better analyze the relationship among the gene structure, conserved motifs, and evolution, a NJ phylogenetic tree consistent with the results in Figure 2 was constructed (Figure 1). The results showed that members of the same subfamily had similar intron numbers and exon–intron structure, which further confirmed the results of apple HD-Zip classification. The average gene length in subfamily III was significantly greater than that in subfamilies I and II. The number of introns in subfamily I and II ranged from zero to four, while subfamily III possessed 18 exons (Figure 3b).

Conserved protein motifs are important in the study of evolution. Three conserved motifs were identified in the *MdHDZ* gene family, their lengths being 29, 41, and 23, respectively (Figure 3 and Appendix A). All MdHDZ proteins had all three motifs, which also showed that the corresponding genes of this category were conserved during evolution. In addition, members of the same subfamily had similar motif locations and distributions. The motif location of subfamily III genes differed from that of subfamily I and II genes, whose motifs were located at the 5′ end of the sequence.

### 2.4. Cis-Regulatory Element Analysis of MdHDZ Genes

To predict the transcription characteristics and gene function of *MdHDZ* genes, cis-regulatory elements were predicted using PlantCARE, based on the 2-kb promoter regions of the genes. Briefly, five hormone-related elements, namely MeJA, SA, GA, ABA, and IAA responsiveness, were identified. Moreover, eight and five putative cis-elements related to plant growth and stress, respectively, were detected. The most frequent cis-acting elements of subfamilies I, II, and III were ABA-responsive elements, MeJA-responsive elements, and elements essential for anaerobic induction, respectively (Figure 3).

### 2.5. Tissue-Specific Expression Pattern of MdHDZ Genes

For more insight into the potential function of MdHDZs, we screened, in silico, the expression levels of *MdHDZ* genes in 32 organs or tissues (Appendix A) at different apple development stages in the GEO database. In general, the *MdMDZ* genes were constitutively expressed in almost all tested tissues. As shown in Figure 4, the expression patterns of *MdHDZ* genes could be divided into three major groups. Group B had a maximum of 24 genes, whose expression levels were significantly higher in the reproductive organs (flowers, fruits, and seeds) than in the vegetative organs (roots, stems, and leaves). Group C had a minimum of seven genes, whose expression levels were relatively high in vegetative and reproductive organs. Group A had 17 genes, whose expression levels were similar to, but generally lower than, those of group B genes. In brief, our results indicated that MdHDZs belonging to different subgroups may function in different processes of growth, development, and stress response.

### 2.6. Transcriptome Analysis of Apple Leaf Response to AAAP Infection

We analyzed the accumulation of ‘HanFu’ apple leaf transcripts over a 0–48 h period following AAAP inoculation using RNA sequencing (RNA-Seq). Each sequenced sample had 65.61–69.97 million raw reads (Appendix A). After cleaning, 60.88–62.84 and 60.85–62.84 million clean reads belonged to the reference genome (Appendix A) and specific genes (Appendix A), respectively. In total, 78.49–83.15% unique reads were mapped to the reference genome and specific genes. A total of 42,015 gene expression levels in each sample were calculated. After AAAP infection, 46 *MdHDZ* genes were expressed in five stages (i.e., 0, 6, 18, 24, and 48 HPI). Approximately 21% of *MdHDZ* genes showed significant differences, and all belonged to subfamilies I and II (Figure 5a). To validate our RNA-Seq results, 14 *MdHDZ* genes that showed significant differences were selected for qRT-PCR. The qRT-PCR results of these genes in five samples were consistent with our RNA-Seq data (Figure 5b).

Furthermore, DEGs between the AAAP-infected and control samples were detected. A total of 10,156, 10,442, 10,413, and 12,791 DEGs were identified when 6 vs. 0, 18 vs. 0, 24 vs. 0, and 48 vs. 0 HPI libraries were compared, respectively (Figure 6a). Across all comparisons, 4217 DEGs were common, of which 2316 were upregulated and 1901 were downregulated (Figure 6b). These findings indicate that, as the disease progressed after AAAP infection, a great change was observed in the gene transcription levels of ‘HanFu’ apple leaves. According to the expression, we clustered DEGs with the same expression pattern into 12 clusters related to the five stages. Seven differentially expressed *MdHDZ* genes were placed in four clusters, namely *MdHDZ22* in cluster 2; *MdHDZ35* in cluster 3; *MdHDZ2*, *MdHDZ4*, *MdHDZ16*, and *MdHDZ34* in cluster 8; and *MdHDZ9* in cluster 11. In cluster 2, gene expression decreased rapidly at six HPI, and was maintained. Gene expression in cluster 5 was significantly upregulated at six HPI, and then down regulated at 18 HPI. In cluster 8, gene expression increased continuously at six HPI, while in cluster 11 it increased continuously at six HPI and was maintained (Figure 6c).

To predict the functions of *MdHDZ* genes in response to AAAP infection of apples, GO enrichment analysis of DEGs was performed in clusters 2, 3, 8, and 11. DEGs in cluster 2 were mainly enriched in sulfate reduction, protein import into the mitochondrial intermembrane, and response to carbon dioxide (Figure 7a). DEGs in cluster 3 were mostly involved in beta-amylase activity (Figure 7b). DEGs in cluster 8 were mainly enriched in NAD(P)H dehydrogenase activity and chitin binding (Figure 7c). DEGs in cluster 11 were mainly involved in chitin binding and cholestenol delta-isomerase activity (Figure 7d).

### 2.7. Expression Profiles of MdHDZ Genes in Adventitious Bud Regeneration from Apple Leaves In Vitro

To predict the role of *MdHDZ* genes in apple adventitious bud regeneration, we downloaded the transcriptome data of our previous study [44]. In that study, we had constructed four RNA libraries using leaves that were 3, 7, 14, and 21 d post inoculation (DPI), in regeneration medium. In addition, we used untreated leaves as a control (0 DPI). Transcriptome analyses were performed on all five RNA libraries through RNA-Seq. Expression levels of 45117 genes were calculated using the FPKM method. Moreover, gene expression of the experimental and control samples was compared (3 vs. 0 DPI, 7 vs. 0 DPI, 14 vs. 0 DPI, 21 vs. 0 DPI).

During apple adventitious bud regeneration, 46 *MdHDZ* genes were expressed in five stages, and eight *MdHDZ* genes were identified as DEGs (fold change ≥ 2; adjusted *p* ≤ 0.001) in all four comparisons (Figure 8a). To validate our RNA-Seq results, all differentially expressed *MdHDZ* genes were selected for qRT-PCR in the current study. The qRT-PCR results of these *MdHDZ* genes, in five samples, were consistent with the RNA-Seq data (Figure 8b).

## 3. Discussion

Plant-specific HD-Zip TFs are considered vital for the regulation of plant growth and tolerance of environmental stresses [1,9,12,34]. Thus far, systematic identification of *HD-Zip* family genes has been pursued in multiple species [24,35,37,38,39,40]. However, a genome-wide analysis of this family in apples has not been conducted. In the present study, we detected 48 *MdHDZ* genes in the apple genome through the synthetic bioinformatics method. The physicochemical properties, gene structure, chromosome distribution, evolutionary relationship, tissue-specific expression level, and expression patterns under AAAP infection and during the regeneration of apple leaves were analyzed in vitro. Our results provide valuable information for the further functional identification of *MdHDZ* genes.

The *HD-Zip* gene family is expanded (48 MdHDZs) in apples compared with that of many species, such as *Arabidopsis* (47 AtHDZs), rice (33 OsHDZs), and potatoes (43 StHDZs). Gene replication, and especially fragment replication, has a significant role in gene family expansion and plant adaptation to environmental changes [45,46]. Roughly 85.42% of *HD-Zip* genes were distributed in duplicated blocks, indicating that fragment duplication is one of the main drivers promoting the expansion of the apple *HD-Zip* gene family. A study demonstrated that chromosome pairs 3 and 11, 9 and 17, and 13 and 16 in apple mainly come from common ancestors [42]. In our analysis, most genes on these chromosomes were located almost in the same position on each corresponding chromosome, which indicated that the chromosomal distribution of genes occurred along with the evolution of apples.

HD-Zip proteins from plants are generally classified into four subfamilies: HD-Zip I, HD-Zip II, HD-Zip III, and HD-Zip IV, in most species [35,37]. Phylogenetic analysis and sequence alignment showed that the *MdHDZ* genes were divided into three subfamilies, which is the same as in citrus plants [36]. The *Arabidopsis* subfamily IV genes are separate, without a homolog in apple and citrus plants, implying a lineage-specific gene loss in these two species. HD-Zip I members in *Arabidopsis* have more diverse compositions than HD-Zip II and III members, as they may be evolutionarily more ancient than HD-Zip II and III members, which allows more time for gene duplication and rearrangement [1,47]. In apples, HD-Zip I is also the most abundant subfamily, which is consistent with previous reports.

The structural characteristics of multiple gene families can reflect their evolutionary trends [48], while the conserved motifs can reflect their protein-specific functions [49]. Here, we found that the gene structure and motif arrangement of MdHDZs in the same subfamily are similar, which indicates that the functions of genes in different MDHDZ subfamilies gradually change during evolution, thereby helping organisms adapt to environmental changes. In terms of gene structure and conserved domains, *HD-Zip* I and II genes are similar, but quite different from *HD-Zip* III genes. Studies on the function of *HD-Zip* I and II genes in *Arabidopsis* have shown that some members of these subfamilies have the same target genes, and play similar regulatory roles in response to pathogen infection [50,51]. Therefore, the functions of HD-Zip I and II genes in apples may be somewhat similar. The gene expression profile is a crucial clue for predicting possible gene functions [52]. No system data on the tissue-specific expression of apple *HD-Zip* genes are available. In the current study, *MdHDZ* genes were constitutively expressed in almost all tested tissues, and displayed three major expression patterns, suggesting their functional diversification during apple growth and tolerance to environmental stresses. In addition, many duplicate gene pairs (e.g., *MdHDZ7* and *MdHDZ14*) were divided into different groups according to tissue-specific expression patterns. This phenomenon may have resulted from the subfunctionalization that may occur between gene pairs.

To date, the functions of many *HD-Zip* genes have been clearly identified in model plants, and members of the same subfamily have similar functions [35,37]. Recent studies have demonstrated that HD-Zip I and II are key regulators of some biotic stresses [5,53,54]. *MdHDZ* gene expression after AAAP infection was explored through RNA-Seq and qRT-PCR. Many *MdHDZ* genes, especially *HD-Zip* I and II genes, were induced in responses, indicating that these genes play a role in the response to fungal infection. Subfamily I genes were strongly induced, and shared high homology with *AtHB13* and *HAHB4*. *AtHB13* overexpression triggered the transcript level of many stress-specific TFs, thus making *Arabidopsis* more resistant to powdery mildew fungi [5]. Heterologous *HAHB4* expression in *Arabidopsis* induces extensive JA synthesis, which can activate the expression of resistance-related genes such as TPI [55]. The potato *HD-ZIP* II genes *StHOX28* and *StHOX30* are rapidly induced under *Phytophthora infestans* stress [38]. Heterologous expression of the capsicum gene *CaHB1,* in tomatoes, enhanced resistance to *P. capsici* by activating *SlPR1*, *SlGluA*, *SlChi3*, and *SlPR23* expression [56]. *MdHDZ9*, *MdHDZ34,* and *MdHDZ35* have high homology with *HD-Zip* II genes of these species, and were instead strongly induced with AAAP infection. Furthermore, many hormone-related elements, such as JA, were found in the promoter region of these apple *HD-Zip* genes. These results indicate that these *MdHDZ* genes may be crucial in apple responses to AAAP infection.

Studies have confirmed that most class III genes are responsible for sustaining the shoot apical meristem [9,10]. Recent studies have demonstrated that class III genes play a vital regulatory role in plant regeneration. For example, *Arabidopsis* HD-Zip III can specifically bind to B-type ARRs to form a transcription complex that activates *WUS*, a key factor regulating bud regeneration [32]. *MxHB13* overexpression can make *M. xiaojinensis* break through age and hormone restrictions and significantly improve adventitious rooting ability [57]. However, the potential regulatory roles of *HD-Zip* genes in adventitious bud regeneration in apples remain unknown; this information would be beneficial for the production of transgenic apples. Therefore, we monitored the expression patterns of *MdHDZ* genes in adventitious bud regeneration from apple leaves. The transcriptional levels of most apple *HD-Zip* III genes significantly increased, indicating that *MdHDZ* III genes might be involved in the regulation of adventitious bud regeneration.

## 4. Materials and Methods

### 4.1. Identification of HD-Zip Genes in The Apple Genome

*Arabidopsis* HD-Zip amino acid sequences were retrieved from TAIR (http://www.arabidopsis.org, accessed on 2 November 2021). This information was employed as a reference to blast [58] apple HD-Zip proteins in the GDR (https://www.rosaceae.org/, accessed on 2 November 2021) and Phytozome2 databases (https://phytozome.jgi.doe.gov/pz/portal.html, accessed on 2 November 2021). All HD-Zip sequences with an e-value of < 1 × 10^−10^ were retained. Furthermore, the tentative HD-Zip sequences were submitted to SMART (http://smart.embl.de/, accessed on 2 November 2021) and NCBI–CDD (https://www.ncbi.nlm.nih.gov/cdd, accessed on 2 November 2021) to confirm the presence of HD and LZ domains. The size, molecular weights, and isoelectric points of apple HD-Zip proteins were predicted using the Expasy ProtParam tool (https://web.expasy.org/protparam/, accessed on 2 November 2021).

### 4.2. Phylogenetic and Synteny Analyses

Using the neighbor-joining (NJ) method with MEGA-7 software [59], an unrooted phylogenetic tree was constructed on the basis of the result of alignments of full-length amino acid sequences of HD-Zip proteins of *Arabidopsis* and apples. The full-length amino acid sequence alignments were performed using the Clustal W program [60]. Bootstrap values were calculated with 1000 replicates. The gene duplication landscape was obtained for synteny analysis using MCScanX [61]. The GDR website was used to locate and assign chromosomes, and chromosome location and synteny relationship were displayed using TBtools [62].

### 4.3. Gene Structure and Conserved Motif Analysis

The exon–intron structure of 48 *MdHDZ* genes was analyzed using GSDS (http://gsds.cbi.pku.edu.cn/, accessed on 4 November 2021), and data were visualized using Tbtools. The conserved motifs of 48 *MdHDZ* genes were analyzed using MEME (https://meme-suite.org/meme/tools/meme, accessed on 4 November 2021), with the maximum number of motifs set to three and the optimum width of the motifs ranging from 6 to 200 [47].

### 4.4. Promoter Analysis

The 2-kb sequences upstream of individual *MdHDZ* genes, defined as promoter sequences, were extracted from the GDR database using TBtools. Cis-acting elements were analyzed referring to PlantCARE (http://bioinformatics.psb.ugent.be/webtools/plantcare/html/, accessed on 4 November 2021) and visualized using TBtools.

### 4.5. Tissue-Specific Expression Pattern Analysis

Tissue-specific expression data of *MdHDZ* genes were originally downloaded from the NCBI GEO database (https://www.ncbi.nlm.nih.gov/geo/, accessed on 5 November 2021) under the accession number GSE42873. The selected gene expression data covered all the major organ systems of the apple (i.e., flowers, fruits, seeds, roots, stems, and leaves). The FPKM values of all genes in the database have been quality controlled. TBtools software was used to transform normalized log2 values and generate the heat map.

### 4.6. Plant Materials and Fungus Inoculation Method

Six-year-old apple (*Malus domestica* cv ‘Hanfu’) plants, grown at the National Germplasm Repository of Apple (Xingcheng, China (40°37′ N, 120°44′ E)), were used in this study. Pathogenic AAAP strains were provided by JunXiang Zhang (Institute of Pomology, Chinese Academy of Agricultural Sciences) and grown on potato dextrose agar medium for 8 d at 23 °C. Apple leaves (length: 4–6 cm) were collected. The spores for inoculation were suspended in deionized water at 3 × 105 colony forming units (CFU)/mL. In each treatment, four symmetrical points on each leaf were injected with a 2 × 10^5^ CFU/mL spore suspension of AAAP; the control group was inoculated with sterile water instead of spore suspension at the same time. The treated leaves were then returned to the culture dishes and maintained at 23 °C under a 16-h light/8-h dark cycle. Next, the samples were retrieved at 0, 6, 18, 24, and 48 h post inoculation (HPI); each timepoint represented an infection stage, with three replicates. All samples were flash-frozen in liquid nitrogen and immediately stored at −80 °C for RNA extraction.

### 4.7. RNA Sequencing and Data Analysis

Total RNA of the samples was extracted using the plant RNA kit (Huayueyang Biotechnology), and then sent to BGI, Shenzhen, China for next-generation sequencing using the BGIseq500 platform. Raw reads were obtained from the sequencing data after base recognition analysis, and clean reads were obtained after filtering out the low-quality reads with > 10% N and contaminated joints. The clean data were compared to the reference genome and gene sequence using the comparison software program HISAT [63] and Bowtie2 [64]. The genome of the ‘Golden Delicious’ apple (GDDH13) and related gene annotations were downloaded from the GDR database. Gene expression levels were calculated on Rsem [65] using the FPKM method, whereas differentially expressed genes (DEGs) were identified using the DEGseq package [66] based on the following criteria: fold change ≥ 2 and a Q-value of ≤ 0.001 (adjusted *p* ≤ 0.001). The raw sequences were deposited in the Sequence ReadArchive database (accession number: PRJNA758843; https://www.ncbi.nlm.nih.gov/sra, accessed on 2 November 2021). The GO enrichment analysis of annotated DEGs was performed using Phyper (https://en.wikipedia.org/wiki/Hypergeometric_distribution, accessed on 2 November 2021) based on the hypergeometric test. Q ≤ 0.05 was used to correct significance levels.

### 4.8. qRT-PCR Validation

Gene-specific qRT-PCR primers were designed using the online Primer-Blast software (Appendix A, https://www.ncbi.nlm.nih.gov/tools/primerblast, accessed on 2 November 2021). cDNA synthesis was completed through the PrimeScript™ RT Master Mix (Perfect Real Time) (TaKaRa). TB Green Premix DimerEraser (TaKaRa) was used as the labeling agent. *MdActin* was used as the internal reference gene [67]. These reactions were performed on a CFX96^TM^ Real-Time System (Bio-Rad Laboratories, Xingcheng, China); the reaction mixture (25 µL) contained 12.5 μL of TB Green Premix DimerEraser, 7.5 mM of forward and reverse primers, and 2 µL of template cDNA. The PCR program was as follows: 95 °C for 1 min, followed by 40 cycles of 95 °C for 5 s, 58 °C for 30 s, and finally 72 °C for 30 s. The 2^−ΔΔ^ method [68] was used to calculate the relative expression levels of the selected genes. Student *t*-test was performed by Graphpad Prism 7 (https://www.graphpad.com/scientific-software/prism, accessed on 2 November 2021). All error bars were standard deviations (SD) from three biological replicates.

## 5. Conclusions

Genome-wide analysis of the HD-Zip family in apples revealed 48 full-length *HD-Zip* genes in the apple genome. These *MdHDZ* genes were classified into three subfamilies through phylogenetic analysis, with homologs from *Arabidopsis*. Synteny analysis revealed that segmental and tandem duplications contributed to HD-Zip family expansion in the apple genome. In terms of gene structure and conserved motif, this gene category exhibited extreme divergence among different subfamilies during the evolutionary process. Analysis of cis-regulatory elements and tissue-specific expression patterns indicated that MdHDZs may play a role in different aspects of apple growth and tolerance to environmental stresses. Moreover, *MdHDZ* gene expression was analyzed in leaves in response to AAAP infection and adventitious bud regeneration from leaves, indicating that the function of these genes is different among different subfamilies. HD-Zip I and II may play a key role in the response to AAAP infection, while HD-Zip Ⅲ is likely involved in adventitious bud regeneration from apple leaves in vitro. These results contribute to a more comprehensive understanding of the function of *HD-ZIP* family genes in apples.

## Figures and Tables

**Figure 1 ijms-23-02632-f001:**
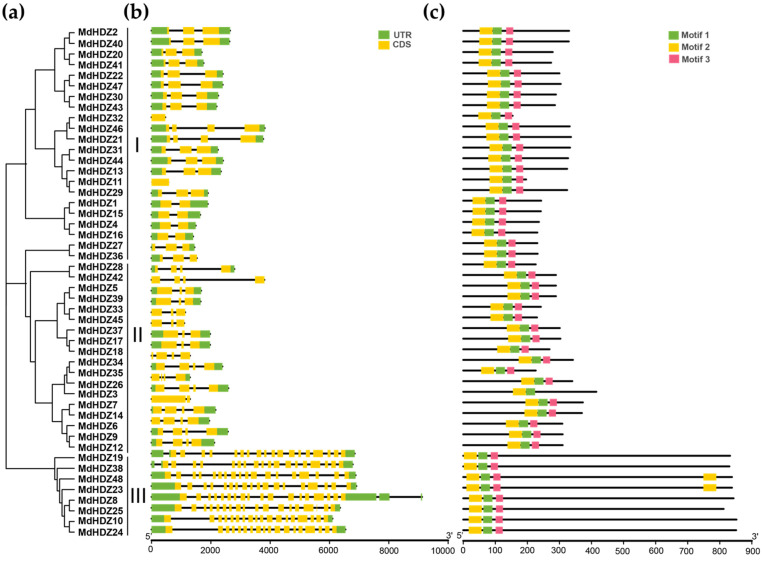
Conserved motif and structures of *MdHDZ* genes. (**a**) Phylogenetic relationship of *HD-Zip genes*. (**b**) Gene structure of *MdHDZ* genes. Green and yellow squares represent UTRs and CDSs, respectively. Black lines represent introns. (**c**) Conserved motifs of *MdHDZ* genes. Green, yellow, and red squares represent motifs 1, 2, and 3, respectively.

**Figure 2 ijms-23-02632-f002:**
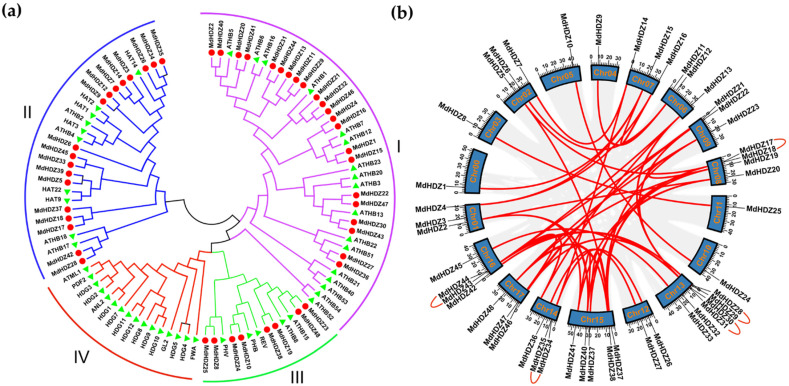
Phylogenetic and synteny analyses of *HD-Zip* genes. (**a**) Phylogenetic relationship of apple and *Arabidopsis HD-Zip* genes. *Arabidopsis HD-Zip* genes are shown by a green triangle, and apple *HD-Zip* genes are shown by a red circle. (**b**) Chromosomal distribution and duplication events of *MdHDZ* genes. Tandem duplicated genes are connected by blue arcs, while segmental duplicated genes are connected by red lines. The scale on the circle is in Megabases.

**Figure 3 ijms-23-02632-f003:**
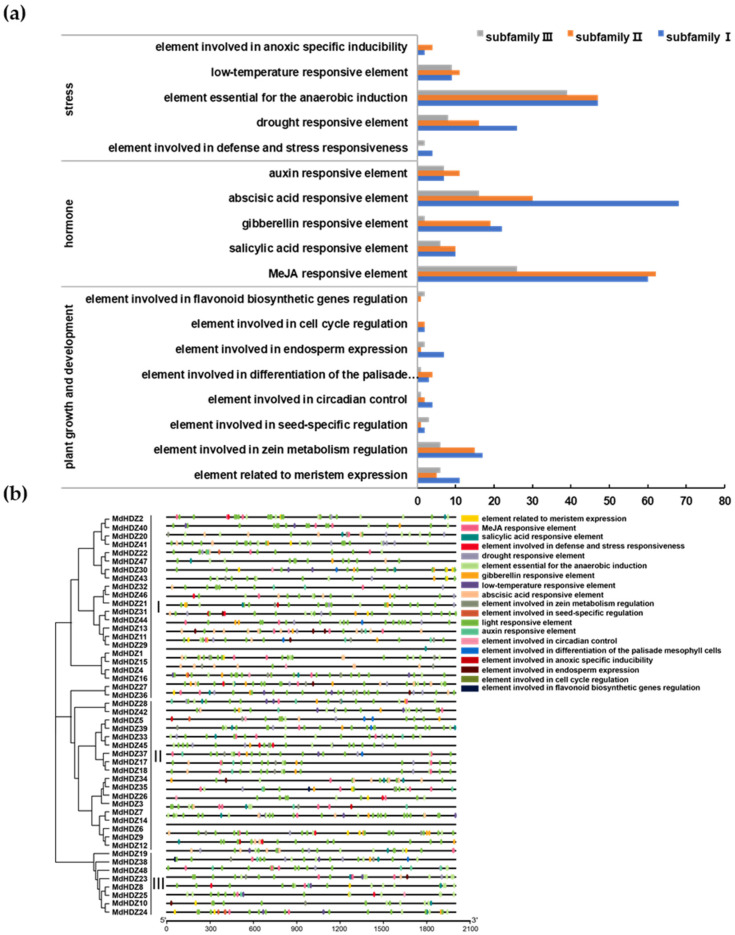
Main cis-acting elements of the 2-kb promoter region of *MdHDZ* genes. (**a**) Numbers and (**b**) chromosomal locations of different types of cis-elements.

**Figure 4 ijms-23-02632-f004:**
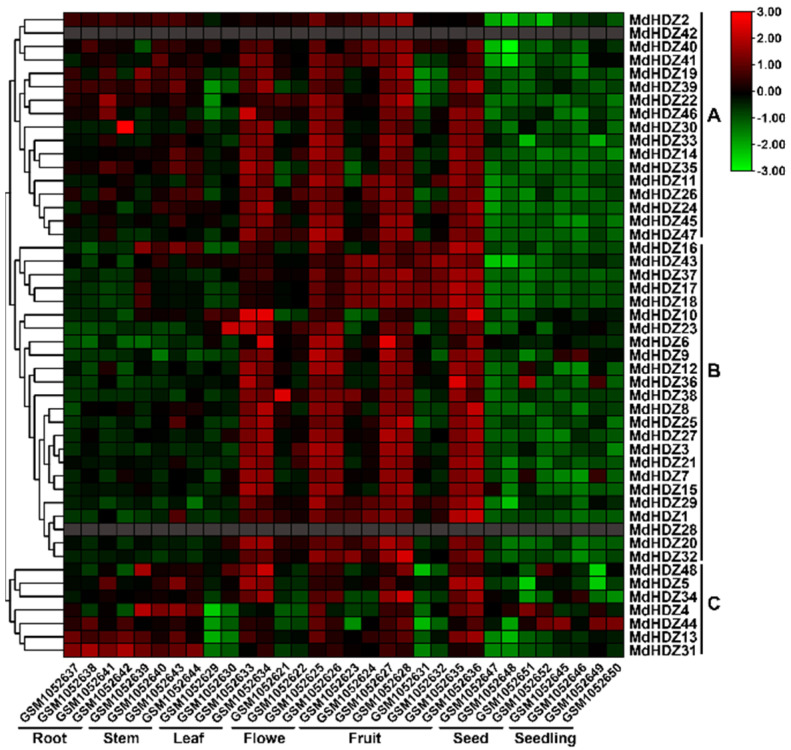
Tissue-specific expression pattern of *MdHDZ* genes. The heat map was generated based on in silico analysis of the tissue-specific expression data of *MdHDZ* genes from the GEO database, and normalized log2 transformed values were used with hierarchical clustering. The transition from green to red represents different expression levels.

**Figure 5 ijms-23-02632-f005:**
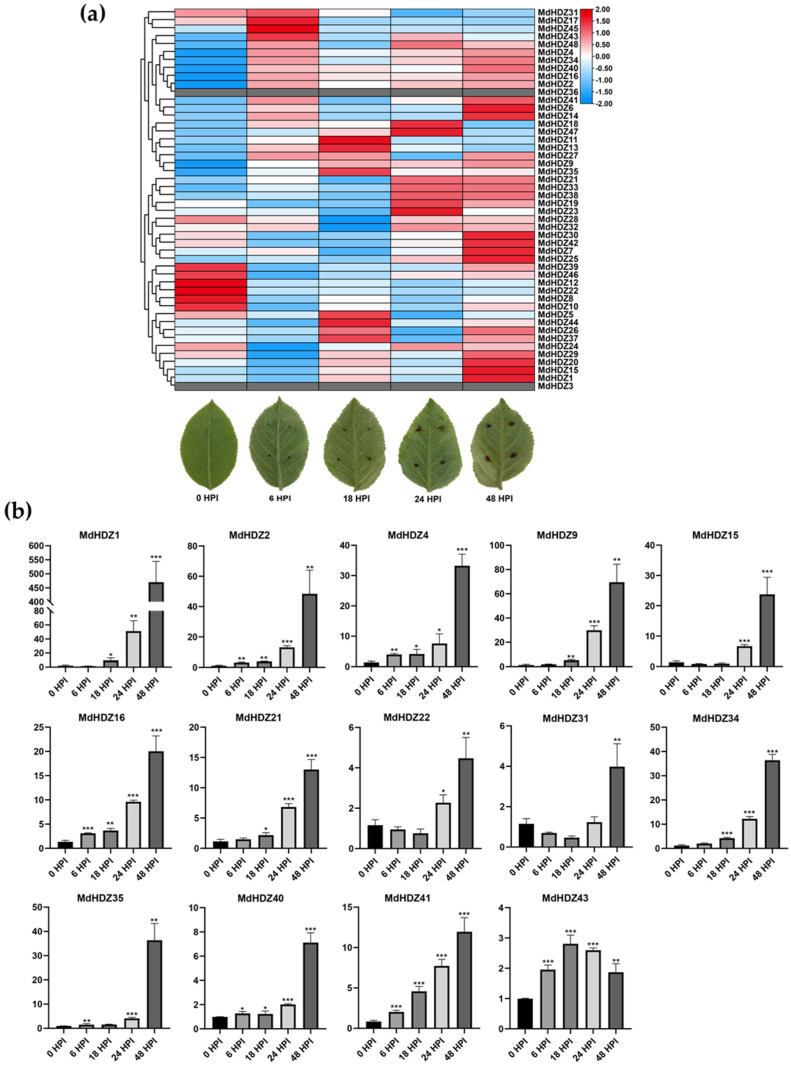
(**a**) Morphological changes and expression pattern of *MdHDZ* genes in response to AAAP infection based on RNA-Seq data. The heat map was generated using TBtools based on the relative expression levels of *MdHDZ* genes in RNA-Seq. Normalized log2 transformed values were used with hierarchical clustering. The transition from blue to red represents different expression levels. (**b**) GO enrichment analysis of DEGs of clusters 2, 3, 8, and 11. (**b**) Expression profiles of 14 selected *MdHDZ* genes in response to AAAP infection, based on qRT-PCR data. Asterisks indicate that the corresponding genes were significantly upregulated compared with the control (* *p* < 0.05, ** *p* < 0.01, *** *p* < 0.001; Student’s *t*-test).

**Figure 6 ijms-23-02632-f006:**
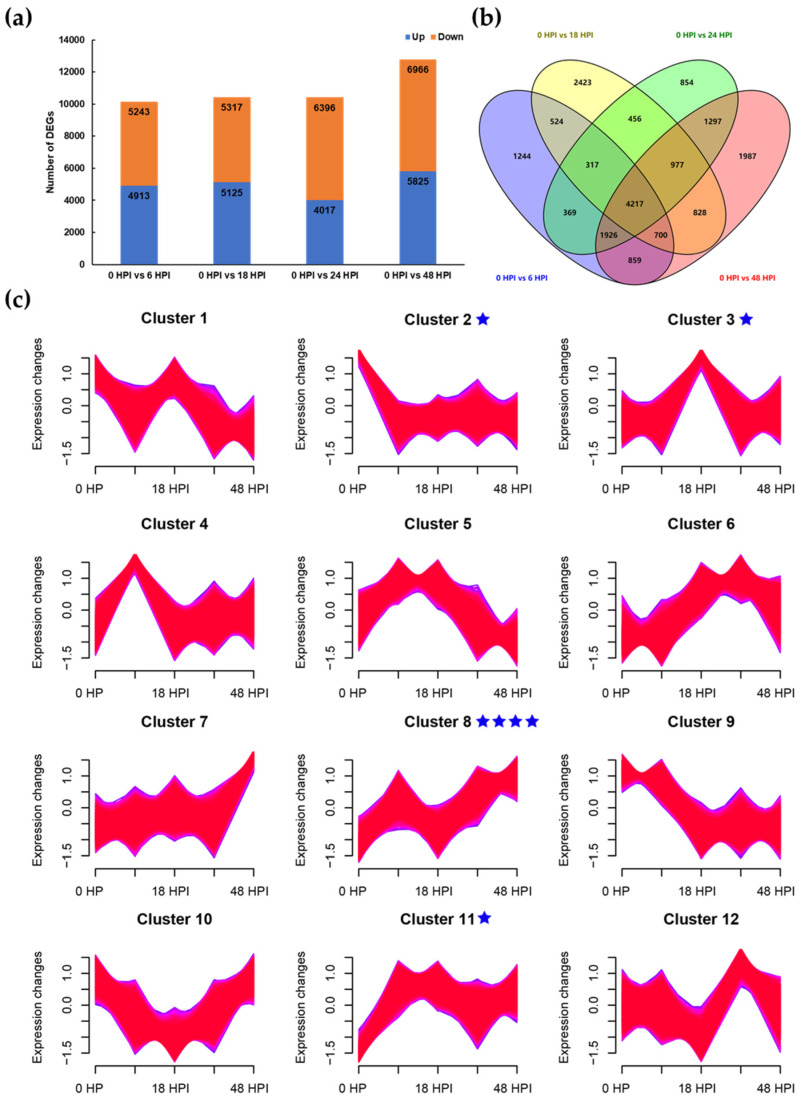
(**a**) Comparisons of DEGs in apples in response to AAAP infection. (**b**) Venn diagram of DEGs in four comparisons. (**c**) Clusters of DEGs and the seven *MdHDZ* genes were labeled with a blue star.

**Figure 7 ijms-23-02632-f007:**
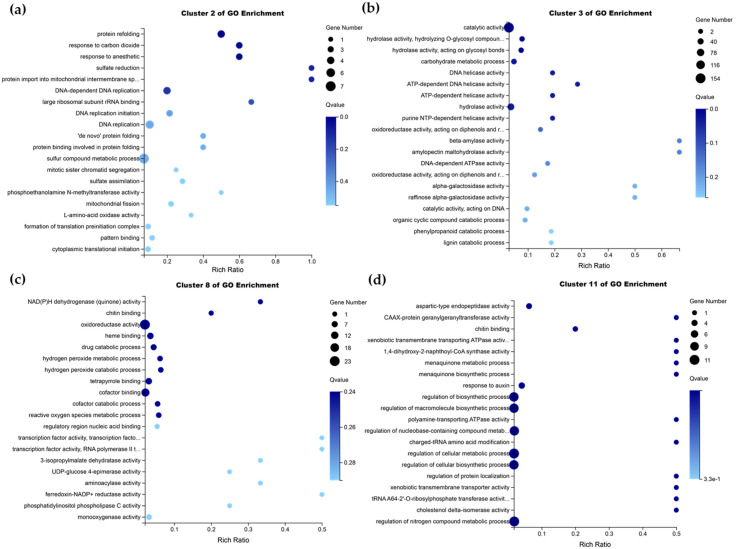
(**a**–**d**) represent the GO functional analysis of DEGs belonging to cluster 2, 3, 8, and 11, respectively. Only the top 15 terms with the smallest Q-value are shown.

**Figure 8 ijms-23-02632-f008:**
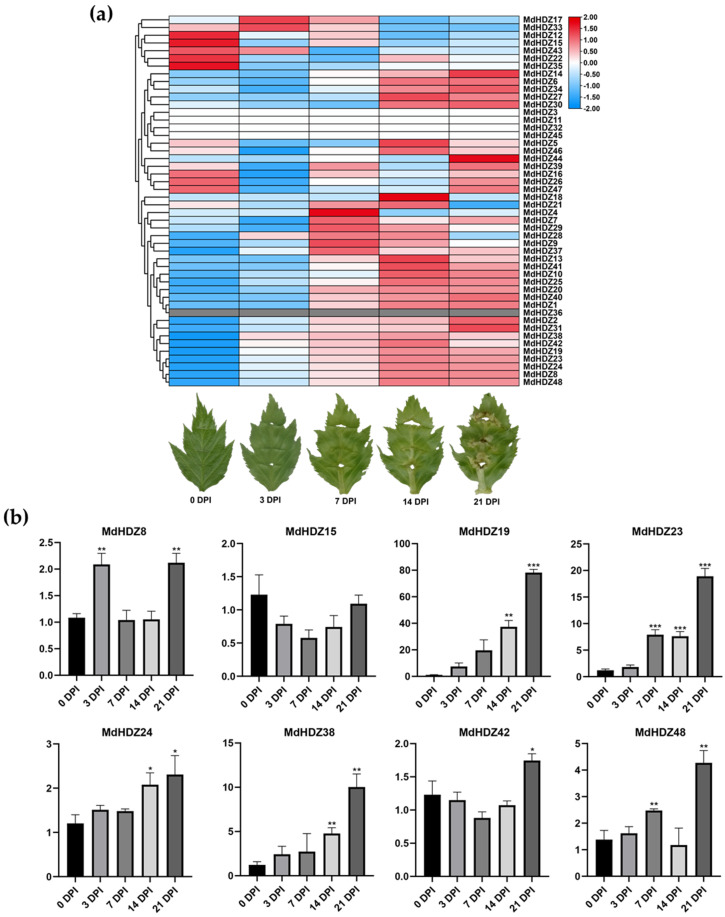
(**a**) Morphological changes and expression pattern of *MdHDZ* genes in adventitious bud regeneration from apple leaves, in vitro, after 3, 7, 14, and 21 DPI in regeneration medium. The heat map was generated using TBtools based on the relative expression levels of *MdHDZ* genes in RNA-Seq. Normalized log2 transformed values were used with hierarchical clustering. The transition from blue to red represents different expression levels. (**b**) Expression profiles of eight selected *MdHDZ* genes expressed during adventitious bud regeneration from apple leaves, in vitro, after 3, 7, 14, and 21 DPI in regeneration medium, based on qRT-PCR data. The relative expression of each gene at 0 DPI was set to 1 for normalization. Asterisks indicate that the corresponding genes were significantly upregulated or downregulated compared with the control (* *p* < 0.05, ** *p* < 0.01, *** *p* < 0.001; Student’s *t*-test).

**Table 1 ijms-23-02632-t001:** Characteristics of *MdHDZ* genes in apples (*Malus domestica* Borkh.).

Gene	Gene ID	Chromosome Location	Exon	Size (Amino Acids)	MW (kDa)	pI
*MdHDZ1*	MD00G1036300	Chr00:6430068..6432722	2	243	27.82	5.37
*MdHDZ2*	MD01G1036200	Chr01:12167559..12171263	3	330	37.26	5.09
*MdHDZ3*	MD01G1069700	Chr01:17372035..17373849	2	415	45.63	8.94
*MdHDZ4*	MD01G1226600	Chr01:31679345..31681430	2	236	27.47	5.36
*MdHDZ5*	MD02G1192800	Chr02:18264264..18266606	3	289	31.93	8.36
*MdHDZ6*	MD02G1216800	Chr02:24770720..24774323	4	309	34.57	8.13
*MdHDZ7*	MD02G1318700	Chr02:37304680..37307701	4	373	41.40	6.06
*MdHDZ8*	MD03G1118500	Chr03:10745381..10758148	18	843	92.52	6.09
*MdHDZ9*	MD04G1061200	Chr04:8078000..8080961	4	310	34.59	7.62
*MdHDZ10*	MD05G1273700	Chr05:40852942..40861482	18	852	92.95	6.03
*MdHDZ11*	MD06G1032300	Chr06:3869510..3870336	1	196	22.40	9.52
*MdHDZ12*	MD06G1054800	Chr06:8046646..8049808	4	310	34.60	8.42
*MdHDZ13*	MD06G1187600	Chr06:32505970..32509240	3	324	36.40	4.79
*MdHDZ14*	MD07G1002500	Chr07:275492..278216	4	370	40.91	6.44
*MdHDZ15*	MD07G1156200	Chr07:22690670..22692966	2	242	27.82	4.75
*MdHDZ16*	MD07G1297100	Chr07:35700128..35702089	2	231	26.66	5.78
*MdHDZ17*	MD08G1075400	Chr08:6123797..6126556	3	303	34.09	6.94
*MdHDZ18*	MD08G1075500	Chr08:6137356..6139170	4	269	30.11	6.60
*MdHDZ19*	MD08G1112900	Chr08:10008344..10017933	18	832	91.75	5.90
*MdHDZ20*	MD08G1188500	Chr08:23799477..23801844	3	279	31.43	5.57
*MdHDZ21*	MD09G1035100	Chr09:2143505..2148775	4	336	38.15	5.29
*MdHDZ22*	MD09G1049000	Chr09:3226829..3230190	3	300	33.92	6.55
*MdHDZ23*	MD09G1205400	Chr09:19576024..19585691	18	838	91.88	6.03
*MdHDZ24*	MD10G1253500	Chr10:34599032..34608186	18	851	93.08	6.03
*MdHDZ25*	MD11G1136800	Chr11:12604948..12613845	18	812	88.65	5.85
*MdHDZ26*	MD12G1055800	Chr12:6297205..6300831	4	340	37.68	8.69
*MdHDZ27*	MD12G1100600	Chr12:15668169..15670196	3	231	27.00	8.37
*MdHDZ28*	MD13G1025200	Chr13:1775080..1778991	4	226	25.40	9.09
*MdHDZ29*	MD13G1030700	Chr13:2201012..2203672	3	324	36.46	4.88
*MdHDZ30*	MD13G1074800	Chr13:5277838..5280985	3	289	32.95	6.06
*MdHDZ31*	MD13G1079500	Chr13:5590336..5593469	3	333	37.30	4.81
*MdHDZ32*	MD13G1196700	Chr13:17146326..17146976	1	154	18.34	9.46
*MdHDZ33*	MD13G1236500	Chr13:23981101..23982690	3	242	27.33	8.02
*MdHDZ34*	MD14G1056200	Chr14:5791182..5794527	4	342	37.78	7.62
*MdHDZ35*	MD14G1056300	Chr14:5813565..5815376	4	226	25.50	8.27
*MdHDZ36*	MD14G1094700	Chr14:14187962..14190102	3	232	27.00	8.95
*MdHDZ37*	MD15G1062900	Chr15:4328848..4331612	3	301	33.96	6.49
*MdHDZ38*	MD15G1092200	Chr15:6404045..6413541	18	830	91.67	5.93
*MdHDZ39*	MD15G1302900	Chr15:29349544..29351859	3	289	31.98	8.12
*MdHDZ40*	MD15G1319800	Chr15:33163160..33166835	3	329	37.18	5.21
*MdHDZ41*	MD15G1374500	Chr15:45816138..45818593	3	274	31.09	6.79
*MdHDZ42*	MD16G1027800	Chr16:1954912..1960238	4	289	32.71	8.99
*MdHDZ43*	MD16G1076000	Chr16:5322782..5325852	3	286	32.52	5.80
*MdHDZ44*	MD16G1079400	Chr16:5571757..5575130	3	327	37.07	4.78
*MdHDZ45*	MD16G1241700	Chr16:26132793..26134339	3	230	26.13	6.83
*MdHDZ46*	MD17G1035400	Chr17:2532964..2538305	4	332	37.43	5.18
*MdHDZ47*	MD17G1049000	Chr17:3572666..3576024	3	304	34.41	6.36
*MdHDZ48*	MD17G1185400	Chr17:22000363..22009994	18	838	92.12	6.14

## Data Availability

The raw sequences of RNA-Seq were deposited in the SRA database in NCBI (accession number: PRJNA758843).

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
