# Peer review of "Genome-Wide Identification, Classification, and Expression Analysis of the HD-Zip Transcription Factor Family in Apple (Malus domestica Borkh.)"

_ijms, 2022, doi:10.3390/ijms23052632_

Round 1

Reviewer 1 Report

The manuscript entitled; Genome-wide Identification, Classification and Expression Analysis of the HD-Zip Transcription Factor Family in Apple (Malus domestica Borkh.) by Liu et al describes about the role HD-Zip transcription factor family against Alternaria alternata. The authors have explained the role of this gene family by combining insilico analysis of protein structures, physiological and biochemical properties, and expression of these genes in three specific tissues. The manuscript is written well but no clear hypothesis was describes to achieve the objectives of current study. I also wonder how authors have confirmed the role of MdHDZ genes in AAAP infection as it is not indicated in the abstract. Further, my comments are as below,

Introduction

Lines 32-36 need further explanation. I will suggest explaining one gene family in one sentence.

A concrete hypothesis is not described. Authors should explain how they will achieve the objective of current study.

Results

Line 104, please replace MdHDZ49 with MdHDZ48.

Materials and Methods

No information is about data analysis. Authors have shown significance of graphs but how they analyzed data. Which software was used?

Reviewer 2 Report

The manuscript ‘ijms-1588271’ deals with very interesting and important topic of genomic study of HD-Zip transcription factors and their role in response to infections in apple. In general, the manuscript is prepared in logical style, with suitable Introduction, valuable Results, very good Discussion and appropriate M&M. Therefore, the manuscript can be conditionally accepted with a subject to Minor revision indicated in the list below. Authors have to address all of the issues but no further involvement of the Reviewer is required.

Minor comments/corrections:

(1) L173-175, Legend of Figure 4 (L186-187) and in other parts. In this study, authors used both own experiments for gene expression and extracted results from Databases. The latter case is typically named ‘in silico’ [in Italics]. Authors have to designate clearly which results were received from own experiments and which data were used ‘in silico’. In this particular paragraph (L173-175), authors mentioned ‘GEO database’ which is good, but their phrase “…we screened the expression..” is confusing because it can mislead readers that these results were generated by the authors of the presented manuscript. To avoid any potential misleading, here and in the beginning of all similar fragments, please insert simple phrase ‘in silico’ or ‘in silico analysis’ or similar, where results were extracted from the existing Databases and used in this study. For example, in L173, it will be as follow: ‘…we screened in silico the expression…’. L186-187: ‘…generated based on in silico analysis of the tissue-specific expression data of…’. Similar, very simple insertions of ‘in silico’ or ‘in silico analysis’ have to be made in other corresponding parts of the manuscript. Please remember that ‘in silico’ has to be written in Italics.

(2) L72, L78. L83, L86, L264, L281, L438 and in other parts. Similar as above, the terms ‘in vitro’ also have to be written in Italics because this is in Latin language. Please check and correct all such cases in entire manuscript.

(3) L56 and L57. Please indicate that AtHB and OsHOX genes are belong to HD-Zip TF genes/ This sentence is started from: ‘From instance…’ but it is not automatically means for which group of genes AtHB and OsHOX are belong to.

(4) L58, L59 and in other parts of the manuscript. This is not compulsory point but I strongly recommend authors to use ‘resistance’ (and ‘susceptibility’) to biotic stresses only like ‘disease resistance’ and ‘resistance to virus’, which is perfectly used in this manuscript. However, in contrast, it is much better to use terms ‘tolerance’ (and ‘sensitivity’) to all abiotic stresses, like in these cases: ‘…tolerance to long-term NaCl stress’ and ‘…performance under salt stress, the drought tolerance of MdHB-7 transgenic…’. Please make corrections.

(5) L65. Could you please correct name of protein ‘GhHB12’ but not ‘Ghhb12’ because it has to be exactly the same as in the encoded gene (L67) and just only in normal case (not in Italics as for the gene).

(6) Table 1. The first column in this Table is headed as ‘Gene’ and it contains names of genes. Therefore, all names in this column MUST be in Italics. The Table capture also has to be clarified. I suppose authors want to say ‘Characteristics of MdHDZ genes in apple (Malus domestica Borkh.)’. If so, the gene names ‘MdHDZ’ and botanical name of the species ‘Malus domestica’ have to be in Italics. Please correct.

(7) Figures 1 and 2. Technical issue. Please insert ‘(a)’ and ‘(b)’ inside of the image and save as ‘Figures. Do not use isolated ‘(a)’ and ‘(b)’ for Figure panels because it makes a trouble as it now in pages 4 and 5.

(8) L133. The name ‘Arabidopsis’ was perfectly used in Italics in entire manuscript. The exception was here in the Figure legend. Please change and use font in Italics for ‘Arabidopsis’ name twice here.

(9) L211. Legend of Figure 5. Please remove “or downregulated” because all results show ‘up-regulation’ or ‘no difference’ compared to the control. Therefore, this phrase can be as follow: ‘…corresponding genes were significantly up-regulated compared to the control…’ or something similar.

Round 2

Reviewer 1 Report

Dear Liu et al. many thanks for sending the revised version. Manuscript is in well shape now and merits publication. Moreover, although most of changes are incorporated however, following point is yet to be tackled.

  1. A concrete hypothesis is not described. Authors should explain how they will achieve the objective of current study.
